# Cow Manure Application Cuts Chemical Phosphorus Fertilizer Need in Silage Rice in Japan

**Thanh Tung Nguyen [1],\*, Yuka Sasaki [1], Mitsuhiko Katahira [1] and Dhirendranath Singh [2]**

1    Faculty of Agriculture, Yamagata University, 1-23 Wakaba-machi, Tsuruoka 997-8555, Yamagata, Japan; yukas@tds1.tr.yamagata-u.ac.jp (Y.S.); mkata43@tds1.tr.yamagata-u.ac.jp (M.K.)
2    The United Graduate School of Agricultural Sciences, Iwate University, Morioka 020-8550, Iwate, Japan; dinosingh19@gmail.com
*    Correspondence: tungctt53@gmail.com

**Abstract:** Cow manure is a good source of phosphorus (P). Here, we investigated whether the amount of P fertilizer can be reduced when cow manure is applied to paddy soil based on growth, P uptake, yield, and soil P status evaluation. Treatments included unfertilized control (CK); manure plus chemical nitrogen (N), potassium (K), and P fertilizer (MNK P); MNK and 75% P (MNK ¾ P); MNK and 50% P (MNK ½ P); MNK and 25% P (MNK ¼ P); and MNK. Manure was applied at the rate of 10 t ha$^{-1}$ in fresh weight base. The P fertilizer was applied at 34.9 kg P ha$^{-1}$ as full dose. Treatment with MNK resulted in the same growth, P uptake, and yield as that with the P fertilizer. P uptake and yield did not respond to P input from chemical fertilizer owing to high soil Olsen P levels. Moreover, MNK could maintain soil Olsen P and total P. Manure application resulted in a positive partial P balance. These results suggest that manure application can cut P fertilizer requirements in P-rich soils, while maintaining soil P for optimal rice growth and yield. By using cow manure in rice production, farmers can conserve finite P resources.

**Keywords:** chemical fertilizer; cow manure; phosphorus balance; Olsen phosphorus; silage rice; yield

## 1. Introduction

The application of chemical phosphorus (P) fertilizer has been routine in agronomic practice since the second half of the twentieth century to increase rice productivity in the intensive agricultural system [1,2]. In Japan, about 30% of the total chemical P fertilizer consumed is used in paddy rice fields at a rate of 40–47 kg P ha$^{-1}$ year$^{-1}$ [3,4]. This application rate is higher than the recommended rate of 20–25 kg P ha$^{-1}$ to maintain the rice yield at 5–6 t ha$^{-1}$ [5]. As a consequence, there is an increase in P-rich soils in paddy fields throughout the country [6,7]. However, Japan is one of the countries with no notable reserves of P for future use [8], and all phosphate rocks used in the manufacturing of chemical P fertilizer are imported [9]. Moreover, these sources of phosphate rocks are finite and presumed to last approximately 370 years at the current rates of exploitation [8,10]. In this context, it is possible, as well as essential, to reduce the use of chemical P fertilizer in Japanese paddy fields. However, this action should be considered based on the evaluation of rice growth, rice yield, and soil P level. Utilization of an alternative P source, such as manure, can be a good management practice to reduce dependance on chemical fertilizers, while maintaining the supply of P to the soil.

Cow manure is considered to be a good source of P for crops [11,12], and an excellent alternative source for chemical P fertilizer. The application of manure in combination with chemical P fertilizer enhances P availability and improves crop yield [13–16]. However, completely replacing chemical P fertilizer with organic fertilizer causes a decline in crop yield [17]. Another concern regarding manure application is that there is a risk of P leaching when excessive amounts of P are applied to the fields. Ohashi [18], Zhao et al. [19], and Qaswar et al. [20] showed that there is a risk of over-fertilization and water-pollution when

chemical fertilization is supplemented with manure. Owing to the above concerns, it is important to consider the effects of manure application combined with different rates of chemical P fertilizer, in maintaining optimal rice growth, rice yield, and soil P level, while reducing the amount of chemical P fertilizer.

Forage rice encompasses rice varieties used to feed animals. The production area of forage rice in Japan has increased in recent times, owing to the reduction in rice consumption and increase in the demand of domestic feed for livestock. From 1960 to 2018, the rice production area in Japan decreased by 50%, from 3.31 to 1.55 million ha [21], and most of the feed for domestic livestock were imported from overseas [22,23]. There are typically two types of forage rice: feed rice and silage rice. Regarding feed rice, only the grains are used as feed for livestock, while the remaining biomass is left on the field. However, in silage rice, all above-ground biomass including panicles, leaves, and stems are harvested, conditioned into silage, and used to feed cattle [22,24], followed by the application of cow manure to the field. This combination is called a mixed rice–livestock system. As manure is an important nutrient input to the paddy field in this system, information on how manure can be substituted for chemical P fertilizer will be an important reference for chemical P fertilizer management in silage rice production areas in Japan.

The objective of this research was to determine the amount of chemical P fertilizer that can be reduced in the silage rice system when manure is applied. To answer this, we investigated the growth, P uptake, yield, Olsen P, total soil P, and partial P balance (PPB) of a silage rice field under cow manure application combined with different levels of chemical P fertilizer.

## 2. Materials and Methods

### 2.1. Study Site

The experiment was conducted at a continuous rice paddy field in the Field Science Center, Faculty of Agriculture, Yamagata University, during the 2020 rice growth season (from May to October). From 1981 to 2010, the average annual precipitation in the study area was 2104.7 mm and the mean annual air temperature was 12.5 °C [25]. The soil type was gleyic fluvisols [26,27]. The soil texture was sandy loam. The initial properties of the topsoil were as follows: bulk density of 1.02 g cm$^{-3}$, pH (water) of 5.5, total carbon (C) of 24.1 g C kg$^{-1}$, total N of 1.82 g N kg$^{-1}$, total P of 954 mg P kg$^{-1}$, available N of 1.61 g NH$_4^+$-N kg$^{-1}$, exchangeable K of 124 mg K kg$^{-1}$, and Olsen P of 57.2 mg P kg$^{-1}$.

### 2.2. Experimental Design

The experiment was performed in a randomized complete block design with three replicates. The treatments consisted of (1) CK (unfertilized control), (2) MNK P (manure plus chemical N, K, and P fertilizer), (3) MNK ¾ P (manure plus chemical N, K, and 75% P fertilizer), (4) MNK ½ P (manure plus chemical N, K, and 50% P fertilizer), (5) MNK ¼ P (manure plus chemical N, K, and 25% P fertilizer), and (6) MNK (manure plus chemical N and K fertilizer). The chemical N, P, and K fertilizers used here were ammonium sulfate, calcium phosphate, and potassium chloride, respectively. Chemical N fertilizer was applied at a constant rate of 140 kg N ha$^{-1}$ in all fertilization treatments. The required N was applied in three splits: 60% as basal application before transplantation, 20% at mid-tillering stage, and 20% at panicle initiation stage. Chemical K fertilizer was applied at the rate of 66.4 kg K ha$^{-1}$ in all fertilization treatments. Chemical P fertilizer was applied at the rate of 34.9 kg P ha$^{-1}$ as full dose in MNK P treatment, and at 26.2, 17.5, and 8.7 kg P ha$^{-1}$ in MNK ¾ P, MNK ½ P, and MNK ¼ P treatments, respectively. Cow manure was applied at the rate of 10 t ha$^{-1}$ in fresh weight base in all fertilization treatments. The manure was composted for 1 year before use from cow dung, remained feed, and rice husk. The manure had almost no smell when it was used. During composting, the microorganisms were killed by the high temperatures. Therefore, the application of this manure is not associated with an increased risk of microbiological contamination of water. The water content of the manure was 53.4% and the nutrient contents in dried weight base were

332 g C kg$^{-1}$, 15.0 g N kg$^{-1}$, 8.00 g P kg$^{-1}$, and 24.3 g K kg$^{-1}$. The nutrient input of manure was calculated by multiplying the nutrient contents to dried weight of manure, resulting in 70.1 kg ha$^{-1}$ of N, 37.3 kg ha$^{-1}$ of P, and 93.9 kg ha$^{-1}$ of K. All manure and chemical P and K fertilizers were applied as a basal application before rice transplantation. There were a total of 18 plots (6 treatments × 3 replications). Each replicate plot area measured 60 m$^2$ (6 × 10 m), and areas were separated by plastic sheets.

### 2.3. Crop Management, Growth Check, and Sampling

The cropping system applied was single-rice crop system. The rice variety used was Fukuhibiki (*Oryza sativa* L., cv. Fukuhibiki), a Japonica type high-yield variety bred by researchers of Tohoku Agricultural Research Center, National Agriculture and Food Research Organization [28]. It was recently recommended as feed and silage rice and is grown in the Tohoku region [29,30]. The experiment was carried out in accordance with the relevant national and international guidelines for Biodiversity Convention [31]. Our group holds full responsibility and authority for the experiments conducted in this field.

The rice was transplanted on 21 May using a transplanting machine. After transplantation, 10 consecutive growth checking hills were selected, starting from the third outermost row from the unplanted border in each plot. The collection of rice growth data, including the tiller number, plant height, and soil-plant analysis development (SPAD) value, was conducted once a week until the heading stage. Measurements of tiller number and plant height were initiated 1 week after transplanting (WAT). The SPAD value was started measuring 2 WAT using a SPAD-502 Plus chlorophyll meter (Konica Minolta Inc., Tokyo, Japan).

Soil was sampled from the topsoil (0–20 cm) at five different positions in each plot at six times: before plowing, 2 WAT, 4 WAT, 6 WAT, 8 WAT, and after harvesting. The soil samples from each plot were thoroughly mixed to make a homogenous composite sample of each replication, dried at 35 °C in a forced-air oven, and then crushed to pass through a 2-mm sieve for chemical analysis. A part of the 2-mm sieved soil sample was finely ground using a grinder (TI-100, Heiko Seisakusho Ltd., Tokyo, Japan) to measure the total C, N, and P. The bulk density of undisturbed soil was determined before the start of the cropping season. This was achieved by collecting soil samples from a depth of 5 cm below the soil surface, using a cutting ring of 50.46 mm inner diameter, 100 cm$^3$ volume, and 50 mm depth [32].

In April 2020, cow manure was sampled and samples were separated into two parts. One part was dried at 60 °C in a forced-air oven, ground finely with a grinder (TI-100) and used for chemical analysis for the calculation of its nutrient input. The other portion was dried at 105 °C in a forced-air oven for the determination of the moisture content of fresh manure, which was then used to calculate the application rate on a dry weight basis.

The rice plants were sampled at mid-tillering, panicle initiation, heading, and harvesting stages. Twelve rice hills were sampled randomly from each plot at three positions. In each of the three positions, four adjacent rice hills were sampled. The rice plants were cut just below the surface to allow some root to remain on the stem, and then were washed to remove all the mud. The remaining root was then cut off from the stem. The stem, leaf, and panicle (in the heading and harvesting stages) were separated and dried at 80 °C in a forced-air oven, ground finely using the grinder (TI-100), and used for chemical analysis. A part of the ground sample was dried at 105 °C in the forced-air oven to analyze the moisture content, which was then used to calculate the amount of dry matter.

Grain yields and yield components were evaluated following the method introduced by Gomez [33]. Straw yield was evaluated as the total air-dried weight of the biomass above ground.

### 2.4. Chemical Analysis

The available P in the soil was determined using Olsen's method [34]: Olsen P was extracted using 0.5 mol L$^{-1}$ NaHCO$_3$ solution with a pH of 8.5 (2.5 g of soil in 50 mL

of solution shaken for 30 min), and the suspensions were filtered. The filtrates were analyzed for Olsen P by the colorimetric measurement of inorganic phosphorus using the molybdate-ascorbic acid method [35]. The available N was determined by anaerobic incubation of air-dried soil at 30 °C for 4 weeks, followed by extraction with 2 M KCl at a soil:KCl ratio of 1:10 ($w/v$). The $NH_4^+$-N content in solution, extracted to measure available N, was determined by steam distillation [36]. The exchangeable K was extracted with 1 M ammonium acetate [37] (pH 7.0) and measured by flame atomic absorption spectrometry (Spectr-AA 220-FS, Varian Australia Pty Ltd., Mulgrave, Australia). The soil pH (water) was determined using a suspension of air-dried soil and water ($w/w$) in the ratio 1:2.5 [38]. Soil total C and total N were analyzed on a Sumigraph NC-220-F analyzer (Sumika Chemical Analysis Service Ltd., Tokyo, Japan). To measure soil total P, the soil was digested with concentrated $H_2SO_4$ plus $H_2O_2$ (30%) at 360 °C [39]. The P concentration in the digested solution was then measured using the vanadomolybdophosphoric acid colorimetric method [40].

To measure the total P and total K in rice plant samples and cow manure, the materials (finely ground samples) were first digested with $H_2SO_4$ plus $H_2O_2$ [41]. The concentration of P in the digested solution was measured using the vandomolybdophosphoric acid colorimetric method [40]. The concentration of K in the digested solution was measured by flame atomic absorption spectrometry (Spectr-AA 220-FS). Total C and total N in cow manure were also analyzed on a Sumigraph NC-220-F analyzer. The nutrient uptake in plants was estimated by multiplying plant nutrient content by dry mass.

### 2.5. Calculation of PPB

PPB is defined as the net change in P content in soil systems that results from accounting for P gains and losses by soil [42]. PPB (kg P ha$^{-1}$ year$^{-1}$) was calculated using the following equation:

$$PPB = \sum P_{inputs} - \sum P_{outputs}$$

where $P_{inputs}$ is the annual P input (kg P ha$^{-1}$ year$^{-1}$) to the field through organic (cow manure) and chemical P fertilizer. $P_{outputs}$ is the P uptake (kg P ha$^{-1}$) by rice plant at the harvesting stage in above-ground biomass.

In the above-mentioned equation, P input from irrigation water and rainfall, and P output through runoff and leaching, were not included, because these amounts were reported to be small and had only negligible effects on the P balance. The P input from irrigation water and rainfall was estimated to be less than 1 kg ha$^{-1}$ in Japanese paddy fields [43–45], whereas the loss of P owing to runoff was estimated to be less than 2 kg ha$^{-1}$ [43,44]. Loss from P leaching is usually insignificant [44,46].

### 2.6. Statistical Analysis

The differences among fertilization treatments for different parameters were analyzed by one-way ANOVA followed by the Tukey–Kramer test at $p = 0.05$ level of significance. The correlations of Olsen P with soil total P, the total P input with total P output and PPB, and the PPB with increases in Olsen P and soil total P were investigated by simple regression analysis. Analysis was performed using the Analysis ToolPak in Excel for Office 365 (Microsoft, Redmond, WA, USA).

## 3. Results

### 3.1. Rice Growth

All fertilization treatments showed significantly higher growth compared with the control (Figure 1). Among fertilization treatments, reduction in chemical P fertilizer application rate showed no significant differences in the number of tillers, plant height, and SPAD values during the entire cropping season. The number of tillers reached the maximum at 35 days after transplanting (DAT) in control, and 42 DAT in fertilization treatments. The plant height increased continuously in all treatments and reached the maximum at 77 DAT. The SPAD values of fertilization treatments increased until about 35 at 28 DAT, and then

decreased a little at 35 DAT before increasing again to reach about 40 at 42 DAT. The SPAD values were then maintained at this high level until 63 DAT before decreasing again.

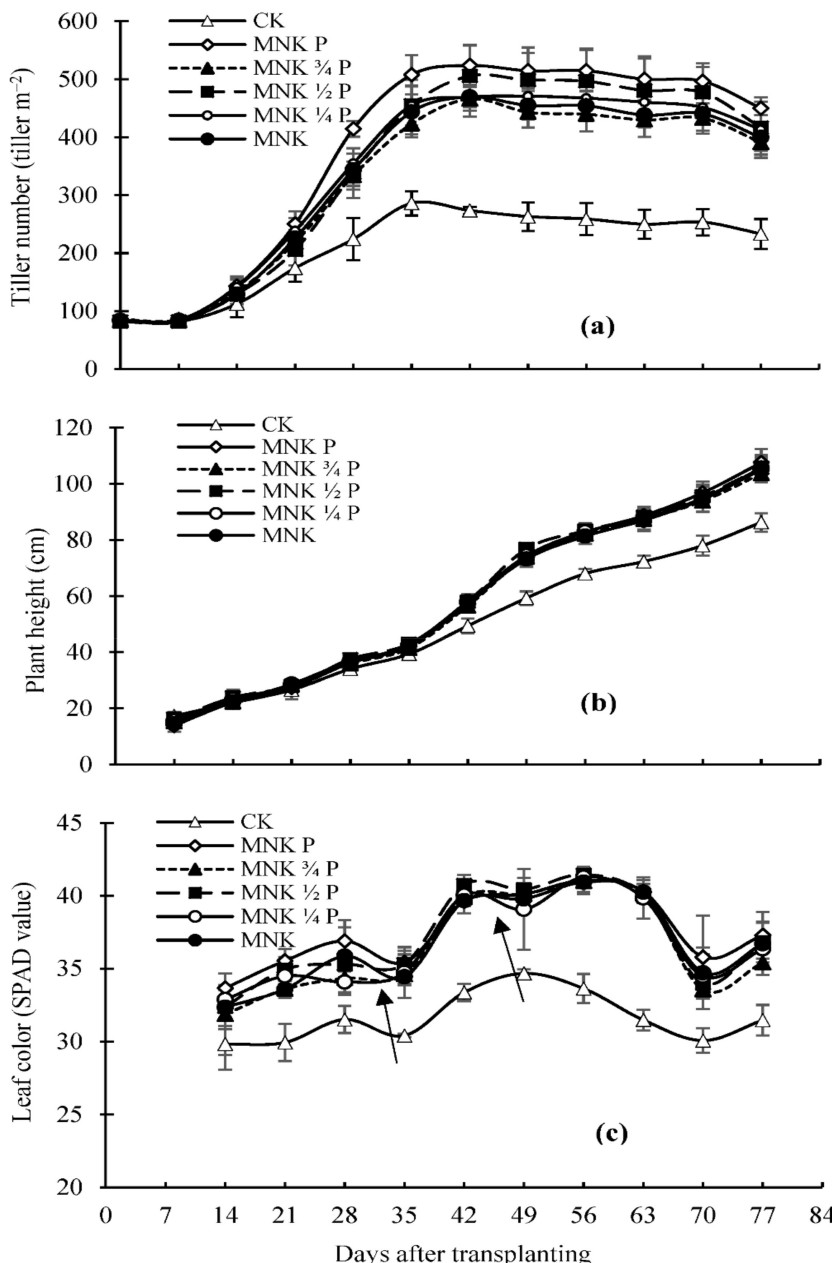

**Figure 1.** Changes in tiller numbers (**a**), plant heights (**b**), and leaf color in SPAD values (**c**) among treatments throughout the experiment. Bars indicate standard deviation (*n* = 3). Arrows indicate the days of chemical N fertilizer application.

### 3.2. Yield Components, Grain Yield, and Straw Yield

The grain yield and straw yield were significantly higher in fertilization treatments than in the control, but showed no significant difference among fertilization treatments (Table 1). The grain yield and straw yield of all fertilization treatments were high and ranged from 7223 to 7768 kg ha$^{-1}$, and from 7672 to 8137 kg ha$^{-1}$, respectively. Regarding yield components, the panicle number was significantly higher in fertilization treatments than in the control. However, the 1000-grain weight in the control was higher than that in the treatments. There were no significant differences in the panicle number and 1000-grain weight among fertilization treatments. The grain number per panicle was higher in MNK

½ P, MNK ¼ P, and MNK than in other treatments. The percentage of unfilled grains was higher in MNK ½ P than in other treatments.

**Table 1.** Yield components, grain yield, and straw yield of rice response to different levels of chemical P fertilizer application.

| Treatments | Yield Components | | | | Grain Yield * kg ha⁻¹ | Straw Yield ** kg ha⁻¹ |
|---|---|---|---|---|---|---|
| | Panicles (No./m²) | Grain (No./Panicle) | Unfilled Grain (%) | 1000-Grain wt * (g) | | |
| CK | 216 ± 5 a | 80.1 ± 8.3 a | 10.7 ± 1.7 a | 25.0 ± 0.1 b | 4207 ± 403 a | 5210 ± 89 a |
| MNK P | 389 ± 23 b | 97.0 ± 3.0 a | 15.3 ± 3.3 a | 23.6 ± 0.2 a | 7569 ± 548 b | 8024 ± 437 b |
| MNK ¾ P | 370 ± 5 b | 95.6 ± 2.1 a | 15.5 ± 2.7 a | 23.4 ± 0.5 a | 7477 ± 122 b | 7888 ± 122 b |
| MNK ½ P | 373 ± 31 b | 108.6 ± 4.1 b | 20.9 ± 5.4 b | 23.6 ± 0.2 a | 7139 ± 537 b | 7672 ± 435 b |
| MNK ¼ P | 378 ± 11 b | 103.1 ± 9.8 b | 17.9 ± 1.9 a | 23.4 ± 0.1 a | 7678 ± 185 b | 8137 ± 202 b |
| MNK | 384 ± 27 b | 98.9 ± 6.7 b | 18.2 ± 3.3 a | 23.5 ± 0.4 a | 7640 ± 405 b | 8081 ± 639 b |

Values are means ($n = 3$) ± standard deviations. Mean followed by different letters within a column indicates significant difference using the Tukey–Kramer test ($p < 0.05$). * 1000-grain wt (weight) and grain yield were brown rice and adjusted to 14% moisture content. ** Straw yield was air-dried weight.

### 3.3. Plant P Uptake

The P uptake in the control was significantly lower than that with fertilization treatments at all stages of sampling (Figure 2), and there were no significant differences among fertilization treatments. The P uptake by leaf and stem increased sharply until the heading stage in all treatments. After this stage, the P uptake was disseminated to the grains resulting in low P content in the leaf and stem. From heading to harvesting, the P uptake was mostly unchanged. At harvest, most of P uptake was by the panicle, with 64.4% in the control and from 73.1 to 74.7% in fertilization treatments. There was no significant correlation between plant P uptake at harvesting and total P input (data not shown).

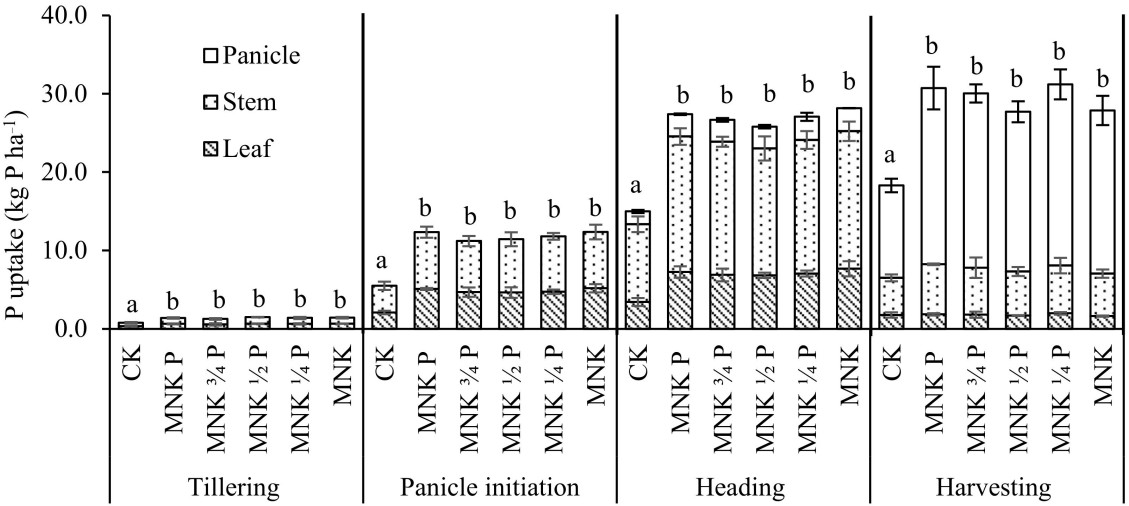

**Figure 2.** P uptake by rice plant and plant parts (leaf, stem, and panicle) at tillering, panicle initiation, heading, and harvesting among treatments. Bars indicate standard deviation ($n = 3$). The different letters in a group indicate significant differences the using Tukey–Kramer test ($p < 0.05$).

### 3.4. Olsen P and Soil Total P

The Olsen P of the soil ranged from 54.8 to 57.8 mg P kg⁻¹ in the plots before the start of the experiment (Table 2). After one cropping season, it increased in all treatments and ranged from 61.7 mg P kg⁻¹ in the control to 68.5 mg P kg⁻¹ in the MNK P treatment. The increasing amounts of Olsen P in the MNK P treatment were significantly higher than those in the control, but did not differ significantly from other fertilization treatments. Moreover,

the increasing amounts corresponded to the level of chemical fertilizer P application in fertilization treatments. After transplantation, soil Olsen P of all treatments decreased slightly, reaching the lowest levels at 4 WAT; thereafter, it started to increase and reached the highest levels at harvesting time. There were no significant differences in Olsen P between treatments at all sampling times.

**Table 2.** Olsen P during cropping season and soil total P response to different levels of chemical P fertilizer application.

| Treatments | Olsen P (mg P kg$^{-1}$) | | | | | | | Total P (mg P kg$^{-1}$) | | |
|---|---|---|---|---|---|---|---|---|---|---|
| | Initial [A] | 2 WAT | 4 WAT | 6 WAT | 8 WAT | After Harvest [B] | (B-A) | Initial [C] | After Harvest [D] | (D-C) |
| CK | 57.7 ± 1.0 a | 50.3 ± 3.6 a | 48.6 ± 1.5 a | 54.9 ± 2.4 a | 53.2 ± 2.8 a | 61.7 ± 2.9 a | 4.0 ± 2.0 a | 950 ± 18 a | 970 ± 8 a | 20.0 ± 19.2 a |
| MNK P | 54.8 ± 3.4 a | 52.8 ± 3.4 a | 55.2 ± 5.1 a | 62.1 ± 5.9 a | 61.5 ± 4.4 a | 68.5 ± 6.6 a | 13.7 ± 3.3 b | 923 ± 16 a | 1013 ± 41 a | 89.5 ± 50.7 a |
| MNK ¾ P | 57.8 ± 1.9 a | 59.0 ± 4.8 a | 55.8 ± 0.5 a | 62.6 ± 7.0 a | 61.9 ± 4.0 a | 66.6 ± 2.0 a | 8.9 ± 2.5 ab | 980 ± 41 a | 1028 ± 35 a | 47.9 ± 6.8 a |
| MNK ½ P | 57.2 ± 3.6 a | 54.7 ± 4.3 a | 53.8 ± 2.5 a | 58.1 ± 1.0 a | 55.1 ± 3.1 a | 65.6 ± 5.0 a | 8.4 ± 4.5 ab | 951 ± 51 a | 1017 ± 10 a | 66.8 ± 59.1 a |
| MNK ¼ P | 57.2 ± 4.0 a | 55.1 ± 5.1 a | 51.7 ± 3.4 a | 55.4 ± 3.0 a | 60.1 ± 0.4 a | 64.3 ± 4.6 a | 7.2 ± 0.9 ab | 963 ± 11 a | 1008 ± 48 a | 45.3 ± 51.9 a |
| MNK | 57.7 ± 2.6 a | 51.6 ± 1.7 a | 53.0 ± 1.9 a | 57.9 ± 2.0 a | 57.0 ± 3.6 a | 64.4 ± 1.3 a | 6.7 ± 3.7 ab | 958 ± 24 a | 1002 ± 13 a | 44.4 ± 25.2 a |

Values are means ($n$ = 3) ± standard deviations. Mean followed by different letters within a column indicates significant difference using the Tukey–Kramer test ($p < 0.05$). (B-A) is the increase in soil Olsen P after one cropping season among treatments. (D-C) is the increase in soil total P after one cropping season among treatments.

The soil total P ranged from 923 to 980 mg P kg$^{-1}$ in the initial soil and from 970 to 1028 mg P kg$^{-1}$ after harvest (Table 2). It increased in all treatments after one cropping season. The level of increase corresponded to the level of chemical fertilizer P application, except in MNP ¾ P (Table 2). However, there were no significant differences in soil total P after harvest and the increasing amounts of soil total P after one cropping season among treatments. A statistically significant correlation between Olsen P and soil total P was observed (Figure 3).

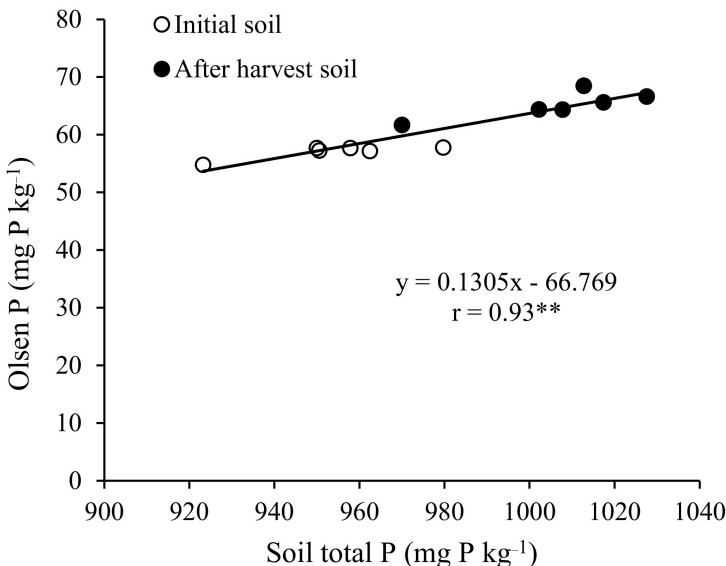

**Figure 3.** Correlation between Olsen P and soil total P of initial soil and soil after harvest of all treatments. **, significant at $p < 0.01$.

### 3.5. PPB

There was no P input in the control, resulting in negative PPB (Figure 4). The P input from manure was higher than total P output by plant uptake, which resulted in positive PPB in all treatments receiving manure. The PPB increased with increasing amounts of chemical P fertilizer application. It also had strong positive correlations with the total P input (Figure 5) and increasing levels of Olsen P and soil total P (Figure 6).

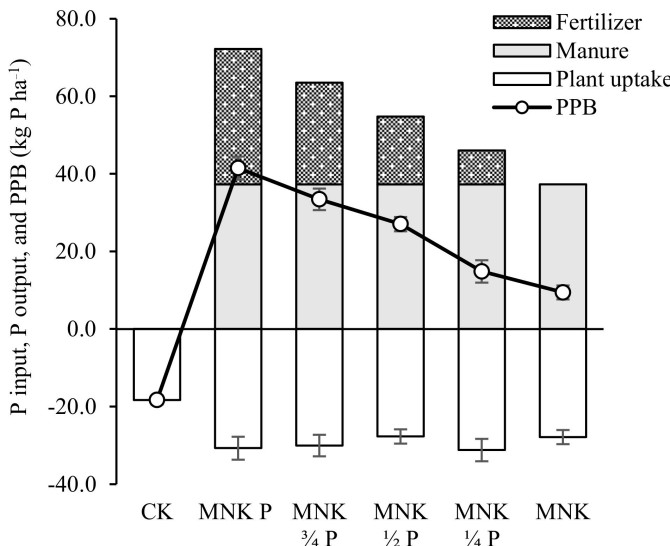

**Figure 4.** The annual P input from fertilizer and manure, P output through plant uptake at harvest, and partial P balance (PPB) among treatments. Bars indicate standard deviation ($n = 3$).

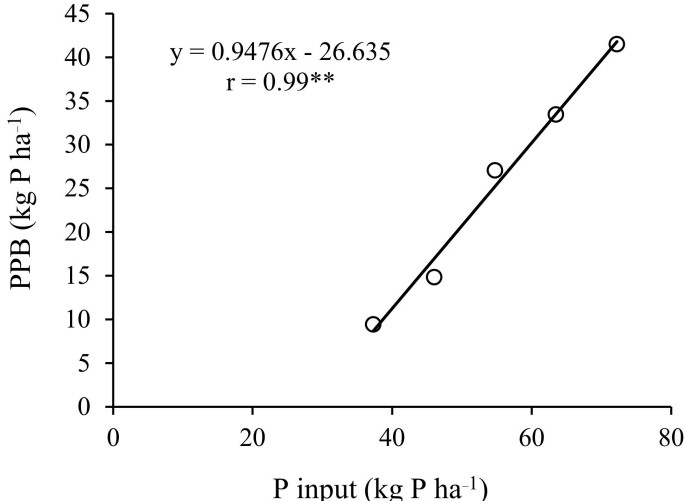

**Figure 5.** The correlation between total P input with partial P balance (PPB) of fertilization treatments. **, significant at $p < 0.01$.

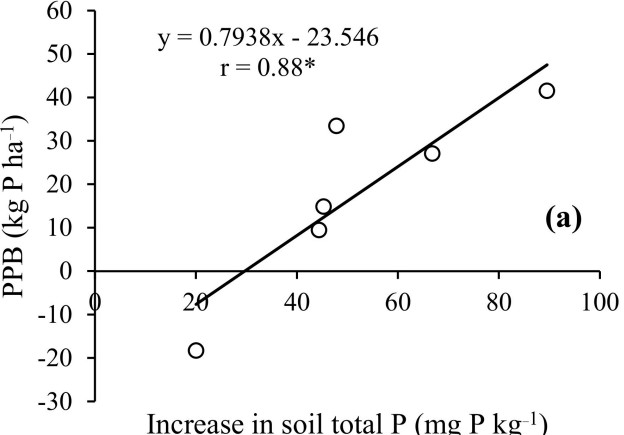

**Figure 6.** *Cont.*

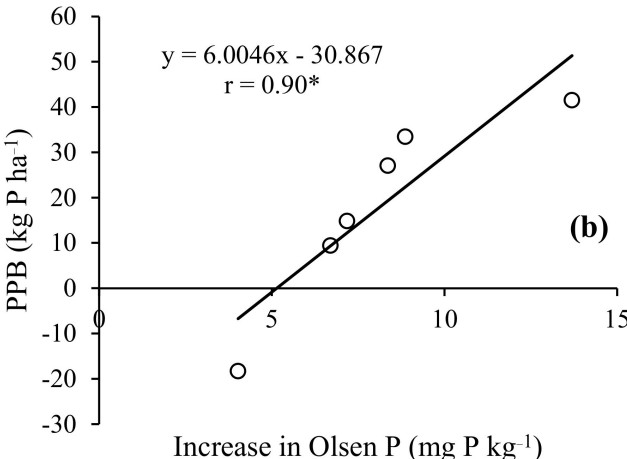

**Figure 6.** The correlation between partial P balance with (**a**) increase in soil total P and (**b**) increase in Olsen P, of all treatments. *, significant at $p < 0.5$.

## 4. Discussion

The P content of cow manure used in this experiment was 8.0 g P kg$^{-1}$, which was higher than that of the manure taken from the same region, as reported by Nguyen et al. [12]. The application of 10 tons of cow manure in fresh weight supplied 37.3 kg P ha$^{-1}$ of total P to the field. This rate of application followed the recommended rate for Yamagata Prefecture (2008), and the amount of P supplied was comparable with that of the national average [4]. Together with chemical P fertilizer application rate at 0, 8.7, 17.5, 26.2, and 34.9 kg P ha$^{-1}$, the total P input to the field ranged from 37.3 to 72.2 kg P ha$^{-1}$. However, the plant growth, plant P uptake, grain yield, and straw yield were not significantly different among fertilization treatments and did not respond to the P input from chemical fertilizer (Figures 1 and 2, and Table 1). This result was attributed to the high levels of Olsen P in our paddy field soils. At the beginning of the experiment, the Olsen P of the field soil ranged from 54.8 to 57.8 mg P kg$^{-1}$ (Table 2), which was much higher than the critical Olsen P level for optimal growth and yield of rice reported by Mallarino and Blackner [47] (11 mg P kg$^{-1}$), Saleque et al. [48] (6–7 ppm), Bado et al. [49] (17 mg P kg$^{-1}$), Zhang et al. [50] (10–20 mg P kg$^{-1}$), and Bai et al. [51] (10.9 mg P kg$^{-1}$). The nonresponse of plant P uptake to P input due to high levels of available P in Japanese paddy field soil was also reported by Nagumo et al. [52]. Therefore, when manure is applied in a mixed rice–livestock system, the application of chemical P fertilizer is not necessary to maintain optimal rice growth and yield. Moreover, P was not a limiting factor to rice growth and yield in our P-rich soils. As the P-rich soil was reported in other Japanese paddy fields [6,7], the results of this study could be applied to other paddy fields in the country.

Information on crop growth and yield is not sufficient to evaluate the sustainability of a nutrient management practice in agricultural systems. For long-term production, it is also important to evaluate the nutrient balance and soil nutrient content. In the case of silage rice, if there is no input of P from cow manure, the PPB of the field will be negative at a level of 19.9 kg P ha$^{-1}$ year$^{-1}$ (this value was calculated based on Figure 5). The negative PPB may not affect the yield in the next season, owing to the rich P soil status; however, long-term continuation of this management practice will cause the soil total P and Olsen P to decrease and become deficient in the soil. There were positive correlations between the PPB and increase in soil total P and Olsen P (Figure 6). A positive correlation between PPB and soil total P content has also been reported in long-term paddy field experiment [53]. Without P input from any source, Olsen P reduced by more than half after 5.6 years, from the initial level of 15 mg P kg$^{-1}$, in paddy soil receiving N and K fertilizer which resulted in a decline in the yield [54]. Therefore, the application of manure in a mixed rice–livestock system is important to maintain positive PPB and the Olsen P levels of paddy soil.

The application of cow manure supplied more P than plant P uptake, resulting in positive PPB (Figure 3). Manure application in this experiment increased total P and Olsen P after one cropping season (Table 2). The application of manure plus chemical P fertilizer increased the total P and Olsen P more than that of only manure application, although there was no significant difference. However, as the growth, P uptake, and yield did not respond to the increase in Olsen P or P input owing to large amounts of Olsen P in the soil, it is not important to increase Olsen P in this P-rich soil. Therefore, the target of P management in a mixed rice–livestock system is to maintain PPB for the maintenance of soil Olsen P levels. The amount of P input required for zero PPB is about 28 kg P ha$^{-1}$ (this value is estimated from the correlation between P input and PPB shown in Figure 5). This result is comparable with that reported by Nagumo et al. [52]. Based on this result, it is also possible to reduce the rate of cow manure application in this silage rice system in terms of balance in P management.

Our results indicate that 100% of chemical P fertilizer application (34.9 kg P ha$^{-1}$) can be reduced, while maintaining optimal rice growth, yield, and soil P level for silage rice in a mixed rice–livestock system. As the total forage rice production area in Japan was 115,000 ha in 2019 [55], it will be possible to reduce the annual chemical P fertilizer application by 4,013,500 kg. Moreover, if this is achieved in all the other rice-producing areas in Japan where rice straw is collected to feed livestock, it will be possible to reduce 55,281,600 kg of chemical P fertilizer application (as total rice production area was 1,584,000 ha in 2019) [55]. Based on the results of this research, we suggest that expanding the mixed rice–livestock system in Japan can reduce the cost for chemical P fertilizer and reserve our finite P sources. However, the results of this research could only apply to P-rich soil paddy fields as those in Japan. Since there are still many P-poor soil paddies over the world, the further studies on how cow manure can substitute chemical P fertilizer in P-poor soil should be conducted to completely understand the advantages of using manure in a mixed rice–livestock system. In addition, the application of cow manure may result in the biological accumulation of pharmaceuticals, including growth hormones and antibiotics, from cow dung into the soil. Further research should be conducted to study these compounds in cow manure, soil, water, and the rice plant to understand the potential impact of cow manure application on the environment and human health.

## 5. Conclusions

Cow manure is the waste of livestock husbandry but serves as a good source of P for crop cultivation. The application of cow manure to paddy fields in a mixed rice–livestock system can supply more P to the field than the amount of P taken up from the field by rice plants, resulting in a positive partial P balance. With regard to the P-rich soil (up to 54.8 mg P kg$^{-1}$) in Japanese paddy fields, application of cow manure alone can supply enough P to achieve optimal rice growth and high yield, as well as maintain the soil P level. The application of cow manure alone increases both soil Olsen P and soil total P. Therefore, by using cow manure, farmers can maintain the P cycle in the rice production system, cut off the dependence on chemical P fertilizers, and conserve finite global P resources.

**Author Contributions:** T.T.N. wrote the main manuscript text and prepared all tables and figures. T.T.N., Y.S., M.K. and D.S. reviewed the manuscript. All authors have read and agreed to the published version of the manuscript.

**Funding:** This research received no external funding.

**Institutional Review Board Statement:** Not applicable.

**Informed Consent Statement:** Not applicable.

**Conflicts of Interest:** The authors declare no conflict of interest.

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
