# Peer review of "Cow Manure Application Cuts Chemical Phosphorus Fertilizer Need in Silage Rice in Japan"

_agronomy, doi:10.3390/agronomy11081483_

Round 1
Reviewer 1 Report
An interesting paper with satisfactory selection of methodologies and valuable data on an important topic. Generally, the results are well presented, however, there are some major comments.
Line 39: Also add the appropriate reference https://doi.org/10.1016/j.catena.2019.104106
Line 78: Provide the climatic data for the experimental site
Line 79: Provide soil classification according to Soil Taxonomy or WRB
Line 79: Provide the soil texture
Line 81: Since rice is one of the few plants that utilize more efficiently NH4+ than NO3- provide the specific forms of available N.
Line 116: Provide the SPAD instrument specifications
Line 147: Justify your choice to use the Olsen method since the pH of soil was < 6.0. It is well known that other methods like Bray is more suitable for acid soils.
Figure 3. The authors should demonstrate only the graph with statistical significant differences
Line 296: The authors should discuss the possible origin of the high amounts of the initial P status of the studied soil and the applicability of this hypothesis for Japan soils.
Conclusions: The authors must highlight the applicability of their results in P – rich soils (up to 50 mg P /kg)
Reviewer 2 Report
In this study, the authors investigated the application of cow manure for agricultural cultivation to cut down the demand for chemical phosphorus fertilizer in Japan. In general, the study has practical significance. The manuscript is well-written and clear. The research methodology is reliable and efficient. The results of the study are encouraging as well as bring benefits to the practice of crop growth.
- The authors should polish more the manuscript regarding grammars and spellings
- The authors should define any abbreviations before use. Some examples include N, P, K, PPB, DAT, SPAD, etc.
- Line 110 page 3. Please, add references for this statement
- Line 111-112 page 3. Revise the sentence ‘As the field…use it’
- Line 160 page 4. Avoid using ‘I, we, you’ in academic writing. Please, use passive voice sentences instead
- Line 164-165 page 4. Revise the sentence ‘The concentration…220-FS’. This information has been mentioned earlier
- Line 168 page 4. Replace ‘with’ by ‘by’ and ‘matter’ by ‘mass’
- As mentioned in the main text, the fresh cow manure was applied to the field. This practice can cause concerns about smells and microbiological contaminants in water/soil. The authors should mention this in the introduction
- The experiments have been triplicated, the authors should show standard deviation values of data when plotting the graphs
- An overall observation from the study that there was no significant difference among fertilization treatments regarding plant P uptake, yield components, grain yield, and straw yield, even till number and plant height due to the P-rich background soil used in the experiments. I suggest that the authors should carry out the tests using infertile soils in further research. As a result, the growth difference among different conditions can be made clear
- Apart from nutrients, other compounds (e.g. hormones, pharmaceutical compounds, antibiotics, etc.) may be present in cow manure. If 100% cow manure is used to cultivate crops, it will cause biological accumulation. The authors should raise this concern in the introduction for further research
Round 2
Reviewer 1 Report
The authors have satisfactorily responded to all my comments and now the paper can be accepted for publication.